# Alcohol consumption increases non-adherence to ART among people living with HIV enrolled to the community-based care model in rural northern Uganda

Norbert Adrawa[1,2,3], John Bosco Alege[3], Jonathan Izudi[3,4]*

1 The AIDS Support Organization, Center of Clinical Excellence, Gulu, Uganda, 2 Department of Business Studies, Faculty of Management Studies, Islamic University in Uganda, Mbale, Uganda, 3 Institute of Public Health and Management, Clarke International University, Kampala, Uganda, 4 Department of Community Health, Faculty of Medicine, Mbarara University of Science and Technology, Mbarara, Uganda

* jonahzd@gmail.com

## Abstract

### Background

Non-adherence to anti-retroviral therapy (ART) is associated with considerable morbidity and mortality among people living with Human Immunodeficiency Virus (PLHIV). Community-based ART delivery model offers a decentralized and patient-centered approach to care for PLHIV, with the advantage of improved adherence to ART hence good treatment outcomes. However, data are limited on the magnitude of non-adherence to ART among PLHIV enrolled to the community-based ART model of care. In this study, we determined the frequency of non-adherence to ART and the associated factors among PLHIV enrolled to the community-based ART delivery model in a large health facility in rural northern Uganda.

### Methods

This analytic cross-sectional study randomly sampled participants from 21 community drug distribution points at the AIDS Support Organization (TASO) in Gulu district, northern Uganda. Data were collected using a standardized and pre-tested questionnaire, entered in Epi-Data and analyzed in Stata at univariate, bivariate, and multivariate analyses levels. Binary logistic regression analysis was used to determine factors independently associated with non-adherence to ART, reported using odds ratio (OR) and 95% confidence level (CI). The level of statistical significance was 5%.

### Results

Of 381 participants, 25 (6.6%) were non-adherent to ART and this was significantly associated with alcohol consumption (Adjusted (aOR), 3.24; 95% CI, 1.24–8.34). Other factors namely being single/or never married (aOR, 1.97; 95% CI, 0.62–6.25), monthly income exceeding 27 dollars (aOR, 1.36; 95% CI, 0.52–3.55), being on ART for more than 5 years

**Data Availability Statement:** All relevant data are within the manuscript and its Supporting Information files.

**Funding:** The authors received no specific funding for this work.

**Competing interests:** The authors have declared that no competing interests exist.

(aOR, 0.60; 95% CI, 0.23–1.59), receipt of health education on ART side effects (aOR, 0.36; 95% CI, 0.12–1.05), and disclosure of HIV status (aOR, 0.37; 95% CI, 0.04–3.20) were not associated with non-adherence in this setting.

## Conclusion

Non-adherence to ART was low among PLHIV enrolled to community-based ART delivery model but increases with alcohol consumption. Accordingly, psychosocial support programs should focus on alcohol consumption.

## Introduction

An estimated 37.9 million people are living with Human Immunodeficiency Virus (HIV) globally [1]. The highest burden of HIV is in sub-Saharan Africa, particularly the East and Southern African region where 20.7 (54.0%) million people were living with HIV by the end of 2019 [2]. HIV is treatable with life-long Anti-retroviral therapy (ART) [3]. The goal of ART is to suppress the level of viral load among people living with HIV (PLHIV) to undetectable levels, to reduce the risk of morbidity and mortality, and to reduce transmission [4]. To achieve these goals, good adherence to ART is critical because it ensures reduced risk of drug resistance, improved overall health, quality of life, and long term survival [4]. Previous epidemiological studies showed that ART adherence is influenced by several factors namely patient, treatment regimen, disease characteristics, healthcare provider relationships, and clinic setting [5–7]. Poor ART adherence tends to result into treatment failure and therefore ART adherence should be routinely assessed and reinforced by every member of the clinical team namely physicians, counselors, nurses, pharmacists, and peer educators amongst others and at all levels of patient care [8].

One approach to addressing the challenges of ART adherence is to ensure the health system provides patient-centered approaches to ART delivery [8], a strategy termed as differentiated service delivery models. A differentiated HIV treatment and care is a strategic mix of approaches to address specific requirements of a subgroup of PLHIV and basically entails approaches that modify client flow, schedules and location of HIV treatment and care services for improved access, coverage, and quality of care. In Uganda, the recommended approaches to differentiated HIV treatment and care include facility and community-based models, both for stable and unstable persons living with HIV [8]. The AIDS Support Organization (TASO) Gulu provides ART through three main models namely health facility, home, and community [9]. Under the community-based ART delivery model, there are designated sites in the community known as community-based drug distribution points where PLHIV come twice or thrice to receive drug refills, clinical evaluation, and psychosocial support [9].

TASO has implemented the community-based drug distribution points approach since 2006 and this approach is considered appropriate in overcoming barriers to retention in care and achieving good rates of viral load suppression [10]. However, this evidence used data from a relatively stable urban setting and the findings may not be generalizable to our setting, a rural and post-conflict region that has suffered at least 2 decades of civil war under the Lord's Resistance Army. In addition, anecdotal evidence reports that some PLHIV enrolled to the community-based ART model of care have failed to suppress their viral load despite ongoing psychosocial support. One of the reasons for failure to achieve viral load suppression is non-adherence to ART. However, data are limited on non-adherence to ART among PLHIV

enrolled to the community-based ART delivery model in our setting. Accordingly, we conducted this study to determine the frequency of non-adherence to ART and the associated patient, clinical, and health systems related factors among PLHIV enrolled to the community-based ART delivery model at TASO Gulu. This study thus provides information that health-care providers, health planners, health policy makers, and health managers amongst others can use to improve the quality of HIV service provision to PLHIV at TASO Gulu and similar settings in developing countries.

## Materials and methods

### Study design and setting

We used an analytic cross-sectional study design since the outcome (non-adherence to ART) and the associated factors (exposures) were assessed at the same point in time. The study setting was TASO Gulu in Gulu district, northern Uganda.

TASO is one of the largest and the first local organizations to respond to the HIV epidemic in sub-Saharan Africa and it has 11 service centers spread across Uganda and 1 training center known as TASO College of Health Sciences located in Kampala, Uganda [11]. Gulu district is located about 335 kilometers by road away from Kampala, Uganda's Capital City. With respect to community-based ART delivery model, there are community drug distribution points, which are sites within the community chosen by PLHIV as being convenient, appropriate, and accessible points for their monthly drug refills. PLHIV who are eligible for community-based ART delivery model are those assessed and deemed stable namely: 1) children, adolescents, pregnant mothers, and adults who have been on their current ART regimen for more than 12 months; 2) those virally suppressed, with viral load less than 1000 copies per ml at the most recent viral load test in the last 12 months; 3) those in the World Health Organization (WHO) clinical stages 1 or 2; 4) those on first or second line ART regimens; and, 5) those with demonstrated good adherence thus over 95% ART adherence in the last 6 consecutive months [8]. For TASO Gulu, there are about 80 community drug distribution points (sites for delivery of community-based ART) spread across the 6 districts of Gulu, Nwoya, Amuru, Pader, Oyam, and Omoro, all in northern Uganda. Elsewhere [12], we have described the setting of TASO Gulu.

### Study population: Eligibility criteria, sample size, and sampling

Eligible participants were PLHIV aged 18 years and beyond enrolled to the community-based ART delivery model. The eligibility criteria for enrollment to the model has already been described under the study setting [8]. Using Yamane's formula shown below [13], we estimated that 381 participants were needed for this study based on the following assumptions: 95% confidence limit, 5% precision (sampling error), and an estimated 8,000 PLHIV enrolled to the community-based ART delivery model.

Sample size (n) = $N/1+N(e)^2$, where N is the total number of PLHIV enrolled to the community-based ART delivery model = 8,000 PLHIV and e is the maximum allowable error = 5% or 0.05. Accordingly, n = $8000/1+8000(0.05)^2$ = 380.9 ≈ 381.

The sample size was distributed proportionally to size of each of the 21 community-based drug distribution points. We conducted sampling in 2 phases. In the first phase, we sampled 21 community-based drug distribution points through simple random sampling approach without replacement to ensure unbiased selection of the study sites. In the second phase, at each of the selected community-based drug distribution points, we used a systematic random sampling approach to establish a sampling interval by dividing the number of participants scheduled for drug refill by the sample size for the study site. From the established sampling

interval, we employed simple random sampling to select a participants until the required number of participants for the site was reached.

## Measurements

The outcome variable was non-adherence to ART measured on binary scale (no or yes), defined according to the Uganda Ministry of Health standard as the number of pills taken divided by the number of pills expected to have been taken, expressed as a percentage. To determine participants who were non-adherent versus those adherent to ART, we used a cut-off of 95%. Participants below this cut-off were considered non-adherent to ART and all the rest were taken as adherent to ART. In addition, we asked the non-adherent participants to provide reasons for missing to take their medications.

The independent variables included age in years, sex, tribe, religion, marital status, level of income in Ugandan shillings and later converted to United States dollars, employment status, level of education, duration on ART, waiting time in hours at the community-based drug distribution point, disclosure of HIV status, years lived with HIV since diagnosis, current consumption of alcohol, receipt of health education on the benefits of ART, current ART side effects, current ART regimen, knowledge of need for life-long HIV treatment; use of reminders to enhance ART adherence, and receipt of counseling in the past 3 months.

## Data collection and quality control measures

Data were collected through researcher administered questionnaire which consisted of both open and closed ended questions in the local language "*Luo*", a predominantly spoken language in the study setting. The questionnaire was forwarded and backward translated, thus from English to *Luo* language and back translated from *Luo* to English language by 2 independent fluent speakers and writers in both languages. The original and translated English versions of the questionnaire were compared and any discrepancies were harmonized by the 2 translators and a final questionnaire was then generated. The final questionnaire was pretested in the neighboring districts of Amuru, Nwoya and Oyam to assess its appropriateness after which some questions were modified appropriately prior to data collection. Data were then collected by research assistants who were trained on the study protocol and supervised by team lead. During data collection, all completed questionnaires were reviewed in real time to ensure data integrity.

## Statistical analysis

Data (S1 File) were single-entered in Epi-Data (Epi-Data Association, Odense, Denmark) [14] impregnated with quality control measures namely skips, range and legal values, and alerts.

We used the Chi-squared test to assess differences in proportions of non-adherence to ART with categorical variables such as sex when the cell count was $\geq 5$ and the Fisher's exact test when the cell count was $< 5$. We used the Student's t-test to asses mean differences in non-adherence to ART with numerical variables like age when data were normally distributed, otherwise the Mann-Whitney U test was used for skewed data. Variables that demonstrated statistical significance at bivariate analysis and those deemed clinically relevant were considered for unadjusted and adjusted binary logistic regression analyses. In the multivariate regression analysis, we included tribe, marital status, level of income, current alcohol consumption, duration on ART, health education on ART related side effects, and HIV status disclosure. We stated the logistic regression analyses results using odds ratio (OR) and 95% confidence interval (CI). The reasons for non-adherence to ART were sorted, categorized, and tabulated, and

then used to triangulate the quantitative findings. The overall analysis was performed in Stata version 15.1 [15], at 5% significance level.

## Ethical issues

Participants who could read and write provided a written informed consent under no due influence of coercion. However, for participants who cannot read and write (those with no formal education), informed consent was obtained through thumb print. Prior to the acquisition of informed consent, the participants were provided information on the purpose of the study, benefits and possible psychological, social, and physical harms if any, and the reasons for their participation in the study. Ethical review and approval was obtained from Clarke International University Research Ethics Committee while administrative approval was obtained from TASO Gulu Institutional Review Board.

## Study reporting

We followed the Strengthening of the Reporting of Observational Studies in Epidemiology (STROBE) guidelines [16, 17] and the ESPACOMP Medication Adherence Reporting Guidelines (EMERGE) [18] to report the study findings.

## Results

### Participants' characteristics

Table 1 shows participant characteristics cross-tabulated by ART adherence. The mean age of all the participants was 42.5 years (standard deviation = 9.7 years). Participants adherent to ART were on average older than those who were non-adherent to ART: 42.5±9.7 versus 40.5 ±9.0 years, respectively ($p = 0.314$). There were statistically significant differences in non-adherence to ART with respect to ethnicity ($p = 0.005$), marital status ($p = 0.004$), household income ($p = 0.020$), duration on ART ($p = 0.036$), and current alcohol consumption ($p<0.001$). We observed no statistically significant difference in non-adherence to ART with regards to sex, educational level, employment status, time taken to reach the community-based drug distribution point, and disclosure of HIV status (all $p>0.05$).

### Level of ART adherence and reasons for non-adherence to ART

The data shows that 25 (6.6%, 93.5%; 4.3–9.5) participants were non-adherent to ART (Table 1). The reasons for non-adherence to ART is shown in Table 2, with the most common reason being forgetfulness, stress, and travels.

### Bivariate analysis of differences in non-adherence to ART

Table 3 shows differences in participant characteristics stratified by non-adherence to ART. No statistically significant difference was observed with respect to years lived with HIV, education on ART benefits and the need for lifelong ART, knowledge of present ART regimen, disclosure of HIV status, receipt of counseling in the past 3 months, and use of reminders to maximize ART adherence (all $p>0.05$).

### Multivariate analysis of factors associated with non-adherence to ART

Table 4 summarized the unadjusted and adjusted analyses results. In the unadjusted analysis, non-adherence to ART was more likely among the non-Acholi (Unadjusted OR (uOR), 3.22; 95% CI, 1.38–752) and single or never married (uOR, 3.91, 95% CI, 1.49–

**Table 1. Characteristics of participants.**

| Participant characteristics | Non-adherent to ART | | | |
| --- | --- | --- | --- | --- |
| | No (n = 356, 93.4%) | Yes (n = 25, 6.6%) | Total (n = 381) | P-value |
| **Age category (years)** | | | | 0.469 |
| < 50 | 268 (92.7) | 21 (7.3) | 289 | |
| ≥ 50 | 88 (95.7) | 4 (4.3) | 92 | |
| Mean ± SD | 42.5±9.7 | 40.5±9.0 | 42.4±9.7 | 0.314 |
| **Sex** | | | | 0.281 |
| Male | 119 (91.5) | 11 (8.5) | 130 | |
| Female | 237 (94.4) | 14 (5.6) | 251 | |
| **Tribe** | | | | 0.005 |
| Acholi | 295 (95.2) | 15 (4.8) | 310 | |
| None Acholi | 61 (85.9) | 10 (14.1) | 71 | |
| **Current marital status** | | | | 0.004 |
| Married | 178 (94.7) | 10 (5.3) | 188 | |
| Single | 41 (82.0) | 9 (18.0) | 50 | |
| Separated | 137 (95.8) | 6 (4.2) | 143 | |
| **Educational level** | | | | 0.473 |
| None | 95 (96.0) | 4 (4.0) | 99 | |
| Primary | 197 (92.1) | 17 (7.9) | 214 | |
| Above primary | 64 (94.1) | 4 (5.9) | 68 | |
| **Employment type** | | | | 0.067 |
| Peasant | 257 (94.5) | 15 (5.5) | 272 | |
| Self | 82 (93.2) | 6 (6.8) | 90 | |
| Formal | 17 (80.9) | 4 (19.1) | 21 | |
| **Income per month (Ugandan Shillings)** | | | | 0.020 |
| ≤100,000 | 273 (95.1) | 14 (4.9) | 287 | |
| >100,000 | 83 (88.3) | 11 (11.7) | 94 | |
| **Duration on ART at time of study** | | | | 0.036 |
| ≤5 years | 179 (90.9) | 18 (9.1) | 197 | |
| >5 years | 177 (96.2) | 7 (3.8) | 184 | |
| **Time to reach the community drug distribution point in hours** | | | | 1.000 |
| ≤ 1 | 102 (93.5) | 7 (6.4) | 109 | |
| >1 | 254 (93.4) | 18 (6.6) | 272 | |
| **Current alcohol consumption** | | | | <0.001 |
| No | 314 (95.4) | 15 (4.6) | 329 | |
| Yes | 42 (80.8) | 10 (19.2) | 52 | |
| **Ever disclosed HIV status** | | | | 0.071 |
| No | 5 (71.4) | 2 (28.6) | 7 | |
| Yes | 351 (93.9) | 23 (6.2) | 374 | |

10.23) participants. Also, income exceeding 100,000 Ugandan shillings (equivalent to 27 dollars) per month (uOR, 2.58; 95% CI, 1.13–5.91) and alcohol consumption (uOR, 4.98; 95% CI, 2.10–11.81) were associated with increased likelihood of non-adherence to ART. On the other hand, non-adherence to ART was less likely when the participant had separated (uOR, 0.78; 95% CI, 0.28–2.20), had been on ART for at least 5 years (uOR, 0.39; 95% CI, 0.16–0.96), had ever received health education about ART side effects (uOR, 0.24; 95% CI, 0.09–0.61), and had disclosed his/her HIV status (uOR, 0.16; 95% CI, 0.03–0.89).

**Table 2. Reasons for non-adherence to ART among patients.**

| Reasons | Frequency | Percentage |
|---|---|---|
| Forgetfulness | 45 | 51.7 |
| Gender based violence | 3 | 3.4 |
| Ran out of pills | 4 | 4.6 |
| Life stresses | 12 | 13.8 |
| On safari (travel to somewhere) | 9 | 10.3 |
| Medication fatigue | 5 | 5.7 |
| Admitted in hospital | 5 | 5.7 |
| Lacked transport to come for refill | 4 | 4.6 |

In the adjusted analysis, non-adherence to ART was significantly associated with alcohol consumption (Adjusted odds ratio (aOR), 3.24; 95% CI, 1.26–8.34). However, non-adherence to ART was not significantly associated with being a non-Acholi (aOR, 1.49; 95% CI, 0.52–4.27), single or never married (aOR, 1.97; 95% CI, 0.62–6.25) or separated (aOR, 0.76; 95% CI, 0.26–2.26), having monthly income exceeding 100,000 Ugandan shillings (aOR, 1.36; 95% CI, 0.52–3.55), being on ART for more than 5 years (aOR, 0.60; 95% CI, 0.23–1.59), ever receiving

**Table 3. Bivariate analysis of differences in non-adherence to ART among PLHIV enrolled to community-based ART delivery model with health services related factors.**

| | Non-adherent to ART | | | |
|---|---|---|---|---|
| Participant characteristics | No (n = 356, 93.4%) | Yes (n = 25, 6.6%) | Total (n = 381) | P-value |
| **Years alive since diagnosed with HIV** | | | | 0.849 |
| ≤3 years | 52 (92.9) | 4 (7.1) | 56 | |
| >3 years | 304 (93.5) | 21 (6.5) | 325 | |
| Mean ± SD | 7.8±3.40 | 6.9±3.25 | 7.8±3.39 | 0.197 |
| **Educated on benefits of taking ARVs** | | | | 1.000 |
| No | 5 (100.0) | 0 (0.0) | 5 | |
| Yes | 351 (93.3) | 25 (6.7) | 376 | |
| **Knows ART is lifelong** | | | | 0.612 |
| No | 14 (100.0) | 0 (0.0) | 14 | |
| Yes | 342 (93.2) | 25 (6.8) | 367 | |
| **Knows his/her current regimen** | | | | 0.176 |
| No | 150 (91.5) | 14 (8.5) | 164 | |
| Yes | 206 (94.9) | 11 (5.1) | 217 | |
| **Attitude of health workers** | | | | 0.422 |
| Receptive | 349 (93.6) | 24 (6.4) | 373 | |
| None receptive | 7 (87.5) | 1 (12.5) | 8 | |
| **Perceived waiting time at the community drug distribution point** | | | | 0.154 |
| Short/just okay | 318 (94.1) | 20 (5.9) | 338 | |
| Long/longer | 38 (88.4) | 5 (11.6) | 43 | |
| **Received counseling in the past 3 months** | | | | 0.750 |
| No | 275 (93.2) | 20 (6.8) | 295 | |
| Yes | 81 (94.2) | 5 (5.8) | 86 | |
| **Use reminders for ART adherence** | | | | 0.652 |
| No | 99 (92.5) | 8 (7.5) | 107 | |
| Yes | 257 (93.8) | 17 (6.2) | 274 | |

**Table 4. Results of logistic regression analysis of factors associated with non-adherence to ART among people living with HIV enrolled to community-based ART delivery model.**

| Participant characteristics | Binary logistic regression analysis | | | |
| --- | --- | --- | --- | --- |
| | Unadjusted analysis | | Adjusted analysis | |
| | OR | 95% CI | aOR | 95% CI |
| **Tribe** | | | | |
| Acholi | 1 | | 1 | |
| None Acholi | 3.22** | (1.38,7.52) | 1.49 | (0.52,4.27) |
| **Current marital status** | | | | |
| Married | 1 | | 1 | |
| Single | 3.91** | (1.49,10.23) | 1.97 | (0.62,6.25) |
| Separated | 0.78 | (0.28,2.20) | 0.76 | (0.26,2.26) |
| **Income per month (Ugandan Shillings)** | | | | |
| ≤100,000 | 1 | | 1 | |
| >100,000 | 2.58* | (1.13,5.91) | 1.36 | (0.52,3.55) |
| **Current alcohol consumption** | | | | |
| No | 1 | | 1 | |
| Yes | 4.98*** | (2.10,11.81) | 3.24* | (1.26,8.34) |
| **Duration on ART at time of study** | | | | |
| ≤5 years | 1 | | 1 | |
| >5 years | 0.39* | (0.16,0.96) | 0.60 | (0.23,1.59) |
| **Educated on ART side effects** | | | | |
| No | 1 | | 1 | |
| Yes | 0.24** | (0.09,0.61) | 0.36 | (0.12,1.05) |
| **Disclosed HIV status** | | | | |
| No | 1 | | 1 | |
| Yes | 0.16* | (0.03,0.89) | 0.37 | (0.04,3.20) |

**Note**: 1) * $p < 0.05$,

** $p < 0.01$,

*** $p < 0.001$ at 5% significance level; 2) All odds ratios (OR) are exponentiated with the 95% confidence intervals in brackets.

health education on ART side effects (aOR, 0.36; 95% CI, 0.12–1.05), and disclosure of HIV status (aOR, 0.37; 95% CI, 0.04–3.20).

## Discussion

This study determined the prevalence of non-adherence to ART and the associated factors among PLHIV enrolled to the community-based ART delivery model in northern Uganda. Our data shows that 6.6% of PLHIV are non-adherent to ART and alcohol consumption increases ART non-adherence. The prevalence of non-adherence to ART in this study is slightly lower than the prevalence observed in previous observational studies [19–21] but similar to another previous study in Uganda [22]. Our findings suggest that a small proportion of PLHIV enrolled to the community-based ART delivery model did not achieve the desired perfect adherence to ART of ≥95% [23]. The community-based ART delivery model therefore appears to have overcome most of the barriers to ART adherence due to improved access to HIV care to ≤5 km radius to one's homestead. The model has made ART delivery more efficient and effective for both the health system and PLHIV. Besides this, the model provides appropriate support to PLHIV that promotes their long-term retention [9]. However, there is also the possibility that the low prevalence of non-adherence to ART is a reflection of enrolling

participants who were likely to be adherent to ART. This further implies that non-adherence to ART cannot be ignored even amongst PLHIV who have proven record of good adherence to ART. Accordingly, strengthening adherence to ART through measures such as treatment supporter, continuous counseling on benefits of perfect ART adherence, use of personal reminders such as alarms and calendars among others is important. One of the main reasons cited for non-adherence to ART is forgetfulness and this is consistent with previous studies [24, 25]. The other reasons included shortage of pills, transportation challenges, and gender-based violence. Previous study medication fatigue [26], which is inconsistent with the reasons reported in this study.

The finding that alcohol consumption increases the likelihood of non-adherence to ART is consistent with several studies elsewhere [27–29]. Indeed, alcohol consumption is a frequent problem among PLHIV [30, 31], more so among those on long term ART (≥6 years).Alcohol consumption has been reported to contribute to substantial number of deaths among PLHIV [32]. Our finding is thus not surprising because alcohol has a disinhibitory effect on cognitive functioning thus disrupting an individual's normal behavior and reasoning. Alcohol consumption is a central cause of forgetfulness, non-adherence to clinical and counseling advice, and disruption of normal social life among others. These effects compromise adherence to ART. It is important to recognize that in our setting, many households brew alcohol to earn a living and access to alcohol is unrestricted provided one is able to buy. Accordingly, with no healthy public policies to regulate alcohol consumption in place, many PLHIV continue to access and drink alcohol. The community-based ART program therefore needs to strengthen the ongoing counseling to focus on the dangers of alcohol consumption namely possible interactions with anti-retroviral drugs, physical and mental health damage, and adverse socio-economic consequences such as loss of income and unemployment among others.

Our study has several strengths and limitations to consider. In terms of strengths, this is one of the few studies to underscore non-adherence to ART among PLHIV enrolled to the community-based ART delivery model in northern Uganda. We assessed non-adherence to ART across a representative sample of community drug distribution points, making these results generalizable to similar settings in Uganda and elsewhere. We used the Uganda Ministry of Health standard definition to distinguish adherence from non-adherence to ART. However, one of the limitations in this study is that adherence to ART was assessed through pill counts and the possibility of participants returning with incomplete number of pills during routine visits to the community drug distribution points cannot be ignored.

This study was conducted in a predominantly rural setting and the findings might not apply to an urban setting due to differences in socio-economic profiles and geographical access to health services. Another limitation is that alcohol consumption was measured as a self-reported response so social desirability bias remains a possibility. One study conducted in Southwestern Uganda showed that PLHIV tend to under report alcohol consumption [24]. Besides this, we did not use the alcohol use disorder identification test tool to grade alcohol consumption and to determine the association with non-adherence to ART. Lastly, since we conducted a cross-sectional study, our findings demonstrate association but not causation. Despite these limitations, our findings underscore that near perfect adherence to ART is possible among PLHIV enrolled to community-based ART delivery model in Uganda and possibly in similar settings.

In conclusion, our data shows that a small proportion of PLHIV enrolled to the community-based ART delivery model in northern Uganda are non-adherent to ART, suggesting the model has lessened most of the barriers to perfect ART adherence. However, alcohol consumption increases non-adherence to ART so ongoing psychosocial support is critical. These findings strengthens the evidence base for scaling up community-based ART delivery model in similar settings across developing countries.

## Supporting information

**S1 File. Dataset.**
(DTA)

## Acknowledgments

We thank the entire staff of TASO for providing routine clinical and psychosocial support to people living with HIV. We also thank people living with HIV for accepting to take part in this research. We are indebted to Clarke International University Research Ethics Committee for providing timely and quality review of the study protocol.

## Author Contributions

**Conceptualization:** Norbert Adrawa.

**Data curation:** Norbert Adrawa, John Bosco Alege, Jonathan Izudi.

**Formal analysis:** Jonathan Izudi.

**Investigation:** Norbert Adrawa.

**Methodology:** Norbert Adrawa, John Bosco Alege, Jonathan Izudi.

**Project administration:** Norbert Adrawa.

**Software:** Jonathan Izudi.

**Supervision:** John Bosco Alege.

**Writing – original draft:** Norbert Adrawa, John Bosco Alege, Jonathan Izudi.

**Writing – review & editing:** Norbert Adrawa, Jonathan Izudi.

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
