## [Decision Letter · Decision Letter 0]

9 Oct 2020

PONE-D-20-22867

Alcohol consumption increases non-adherence to ART among people living with HIV enrolled to the community-based care model in rural northern Uganda

PLOS ONE

Dear Dr. Izudi,

Thank you for submitting your manuscript to PLOS ONE. After careful consideration, we feel that it has merit but does not fully meet PLOS ONE’s publication criteria as it currently stands. Therefore, we invite you to submit a revised version of the manuscript that addresses the points raised during the review process.

We look forward to receiving your revised manuscript.

Kind regards,

Tim Mathes

Academic Editor

PLOS ONE

Additional Editor Comments:

- STROBE is just a reporting guideline. So you cannot follow STROBE in “the design, conduct, and analysis of this study” (line 194). Please report all results according STROBE and ESPACOMP (https://www.equator-network.org/reporting-guidelines/espacomp-medication-adherence-reporting-guideline-emerge/).

- It should be described more clearly which predictors you included in the multivariate regression.

Journal Requirements:

2. Please include additional information regarding the survey or questionnaire used in the study and ensure that you have provided sufficient details that others could replicate the analyses. For instance, if you developed a questionnaire as part of this study and it is not under a copyright more restrictive than CC-BY, please include a copy, in both the original language and English, as Supporting Information.  If the original language is written in non-Latin characters, for example Amharic, Chinese, or Korean, please use a file format that ensures these characters are visible.

Reviewers' comments:

Reviewer's Responses to Questions

**Comments to the Author**

1. Is the manuscript technically sound, and do the data support the conclusions?

Reviewer #1: Partly

Reviewer #2: Partly

2. Has the statistical analysis been performed appropriately and rigorously? 

Reviewer #1: No

Reviewer #2: Yes

3. Have the authors made all data underlying the findings in their manuscript fully available?

Reviewer #1: Yes

Reviewer #2: Yes

4. Is the manuscript presented in an intelligible fashion and written in standard English?

Reviewer #1: No

Reviewer #2: Yes

5. Review Comments to the Author

Reviewer #1: 1. I found the authors digressing away from their research questions in the discussion and taking away more value from the study. The discussion should have focused on the effects of alcohol consumption on adherence.

2. Assumptions made in the determination of the sample size are unclear.

3. Yes

4. There are a lot of grammatical areas in the manuscript.

Reviewer #2: 1. Could you check out the entire manuscript for typos and missing words eg line 85?

Introduction

1.The second sentence of the introduction grossly misrepresents facts on the population in the east and Southern Africa.

2. Literature about factors associated with ART non adherence seem to be readily available. The value of this paper could be on how the community based ART delivery model helps to improve adherence to ART or why this model fails to address alcohol as a barrier to ART adherence. This is because previous studies have reported being single/never married, poverty/less income, fatigue/side effects as reasons/factors associated with non adherence to ART. Health education and disclosure as facilitators of ART. Your study maintains disclosure and health education as being associated with ART adherence in addition to turning previously reported barriers of being single/never married, side effects and less income to not being associated with non adherence. Could the previously reported barriers now turned into being not associated with non adherence be attributed to community ART delivery model?

3. The authors term this study as cross sectional though on further reflection, one wonders whether this was really the right study design. The fact that the authors refer to a drug refill prior to data collection and that at the point of data collection when non adherence was determined, leave one wondering whether this was not a cohort study with a retrospective /prospective start. with cross sectional design, the exposure and outcome happen at the same time. For the current study, the exposure which were the drugs were given earlier and the outcome which was non adherence was determined later. this is typical of cohort Alternatively, this could be a case control study design. Participants were recruited by virtual of being cases/non adherent and worked backwards to determine what could have been the exposure. Considering the above, the manuscript will require reanalysis.

4.Abreviations like CDPPS ought to be written out prior to their use see line 120

5. Prior description of TASO Gulu need to be referenced too.

6. It is not clear why authors did not use non adherence to ART in earlier studies in sample size estimation when the subject is heavily researched.

7. How did the 99 participants with zero education give a written informed consent?

8. Isn't low prevalence of none adherence in this study a reflection of the eligibility criteria to start accessing ART through the community based ART delivery model? This brings in focus the study having been carried out among participants that are likely to be adherent to ART. Also this should have been discussed in the discussion section.

9.The discussion section ought to be structured in such a way that reflects how the community based ART delivery model helps to overcome non adherence to ART than the current format that reproduces what we already know.

10. The conclusion too should reflect to what extent does the community based ART delivery model overcomes barriers to ART non adherence.

6. PLOS authors have the option to publish the peer review history of their article (what does this mean?). If published, this will include your full peer review and any attached files.

Reviewer #1: **Yes: **Dr Munyaradzi Madhombiro

Reviewer #2: No

---

## [Author Response · Author response to Decision Letter 0]

15 Oct 2020

Reviewer #1: 

1. I found the authors digressing away from their research questions in the discussion and taking away more value from the study. The discussion should have focused on the effects of alcohol consumption on adherence.

Author’s response: Thank you for this comment. We have revised the discussion and have focused on effects of alcohol consumption on adherence to ART. We have deleted the earlier discussions on non-significant finings. 

2. Assumptions made in the determination of the sample size are unclear. 

Yes

Authors’ response: We have revised the description of sample size estimation and it reads: “Using Yamane’s formula shown below, we estimated that 381 participants were needed for this study based on the following assumptions: 95% confidence limit, 5% precision (sampling error), and an estimated 8,000 PLHIV enrolled to the community-based ART delivery model. Sample size (n) = N/1+N(e)2, where N is the total number of PLHIV enrolled to the community-based ART delivery model = 8,000 PLHIV and e is the maximum allowable error = 5% or 0.05. Accordingly, n = 8000/1+8000(0.05)2 = 380.9 ≈ 381.”

4. There are a lot of grammatical areas in the manuscript.

Authors’ response: We have checked the entire manuscript and have addressed all the typos/grammatical errors.

Reviewer #2: 

1. Could you check out the entire manuscript for typos and missing words e.g. line 85?

Authors’ response: We have critically checked the manuscript for typos and missing words. The revised manuscript now has no missing words and typos.

Introduction

1. The second sentence of the introduction grossly misrepresents facts on the population in the east and Southern Africa.

Authors’ response: We have updated the data on people living with HIV in east and southern Africa using the updated data from the Centers for Disease Control and Prevention. Please refer to reference # 2.

2. Literature about factors associated with ART non adherence seem to be readily available. The value of this paper could be on how the community based ART delivery model helps to improve adherence to ART or why this model fails to address alcohol as a barrier to ART adherence. This is because previous studies have reported being single/never married, poverty/less income, fatigue/side effects as reasons/factors associated with non-adherence to ART. Health education and disclosure as facilitators of ART. Your study maintains disclosure and health education as being associated with ART adherence in addition to turning previously reported barriers of being single/never married, side effects and less income to not being associated with non-adherence. Could the previously reported barriers now turned into being not associated with non-adherence be attributed to community ART delivery model?

Authors’ response: Thank you for this comment. We have revised the discussion section and have now focused on alcohol consumption and non-adherence to ART. In the earlier submission, we had discussed both significant and non-significant results. Now, the latter discussions have been deleted to avoid deviation/or digression. 

3. The authors term this study as cross sectional though on further reflection, one wonders whether this was really the right study design. The fact that the authors refer to a drug refill prior to data collection and that at the point of data collection when non adherence was determined, leave one wondering whether this was not a cohort study with a retrospective /prospective start. With cross sectional design, the exposure and outcome happen at the same time. For the current study, the exposure which were the drugs were given earlier and the outcome which was non adherence was determined later. This is typical of cohort. Alternatively, this could be a case control study design. Participants were recruited by virtual of being cases/non adherent and worked backwards to determine what could have been the exposure. Considering the above, the manuscript will require reanalysis.

Authors’ response: We are grateful for this comment. We did not follow-up a group of individuals initiated on ART, either prospectively or retrospectively. We assessed adherence to ART through pill counts and collected data on factors associated with non-adherence at same point in time. Since both the outcome and exposures were assessed at same point in time, we thought this fits a cross-sectional study design. We have made slight revision in the description of the study design for clarity and it reads: “We used an analytic cross-sectional study design since the outcome (non-adherence to ART) and the associated factors (exposures) were assessed at the same point in time.” We hope this is acceptable now.

4. Abbreviations like CDPPS ought to be written out prior to their use see line 120

Authors’ response: Thank you. We have fully defined this abbreviation and have revised the sentence for clarity and it reads: “For TASO Gulu, there are about 80 community drug distribution points (sites for delivery of community-based ART) spread across the 6 districts of Gulu, Nwoya, Amuru, Pader, Oyam, and Omoro, all in northern Uganda.”

5. Prior description of TASO Gulu need to be referenced too.

Author’s response: We have provided a reference now. Please refer to reference #9.

6. It is not clear why authors did not use non adherence to ART in earlier studies in sample size estimation when the subject is heavily researched.

Authors’ response: Thank you for this comment. We agree that adherence to ART has been researched in some parts of Uganda, predominantly in urban settings. We deemed that estimates in prior studies might not accurately estimate the sample size for our setting. This is because our setting is predominantly rural, post-conflict, and most of the people are socio-economically poor. It was these differences alongside the assumptions that we have described that made us to calculate a sample size specific to our context. We propose to maintain the estimate. Thank you. 

7. How did the 99 participants with zero education give a written informed consent?

Author’s response: We presume the 99 participants referred to here are those participants without formal education as shown in Table 1. For such participants, informed consent was obtained through thumb prints. We have made a slight revision in the ethical issues section to this effect and it reads: “However, for participants who cannot read and write (those with no formal education), informed consent was obtained through thumb print.”

8. Isn't low prevalence of none adherence in this study a reflection of the eligibility criteria to start accessing ART through the community based ART delivery model? This brings in focus the study having been carried out among participants that are likely to be adherent to ART. Also this should have been discussed in the discussion section.

Authors’ response: We have discussed the implication of the eligibility criteria now. The new sentence reads in part: “However, there is also the possibility that the low prevalence of non-adherence to ART is a reflection of enrolling participants who were likely to be adherent to ART. This further implies that non-adherence to ART cannot be ignored even amongst PLHIV who have proven record of good adherence to ART.”

9. The discussion section ought to be structured in such a way that reflects how the community based ART delivery model helps to overcome non adherence to ART than the current format that reproduces what we already know.

Authors’ response: We are grateful for this comment. We have now focused the discussion on the role of community-based ART in overcoming barriers to adherence. We have removed redundant discussions.

We have include a new sentence that reads: “The community-based ART delivery model therefore appears to have overcome most of the barriers to ART adherence due to improved access to HIV care to ≤5 km radius to one’s homestead. The model has made ART delivery more efficient and effective for both the health system and PLHIV. Besides this, the model provides appropriate support to PLHIV that promotes their long-term retention.” 

10. The conclusion too should reflect to what extent does the community based ART delivery model overcomes barriers to ART non adherence.

Authors’ response: We have incorporated this change in the revised manuscript. The revised conclusion reads: 

“In conclusion, our data shows that a small proportion of PLHIV enrolled to the community-based ART delivery model in northern Uganda are non-adherent to ART, suggesting the model has lessened most of the barriers to perfect ART adherence. However, alcohol consumption increases non-adherence to ART so ongoing psychosocial support is critical. These findings strengthens the evidence base for scaling up community-based ART delivery model in similar settings across developing countries.”

---

## [Editor Report · Decision Letter 1]

23 Oct 2020

PONE-D-20-22867R1

Alcohol consumption increases non-adherence to ART among people living with HIV enrolled to the community-based care model in rural northern Uganda

PLOS ONE

Dear Dr. Izudi,

Thank you for submitting your manuscript to PLOS ONE. After careful consideration, we feel that it has merit but does not fully meet PLOS ONE’s publication criteria as it currently stands. Therefore, we invite you to submit a revised version of the manuscript that consideres the previously raised additional editor comments: 

- STROBE is just a reporting guideline. So you cannot follow STROBE in “the design, conduct, and analysis of this study” (line 194). Please report all results according STROBE and ESPACOMP (https://www.equator-network.org/reporting-guidelines/espacomp-medication-adherence-reporting-guideline-emerge/).

- It should be described more clearly which predictors you included in the multivariate regression.

The original revision should be supplemented by these suggestions. 

We look forward to receiving your revised manuscript.

Kind regards,

Tim Mathes

Academic Editor

PLOS ONE

---

## [Author Response · Author response to Decision Letter 1]

23 Oct 2020

Editor comments:

1. STROBE is just a reporting guideline. So you cannot follow STROBE in “the design, conduct, and analysis of this study” (line 194). Please report all results according STROBE and ESPACOMP (https://www.equator-network.org/reporting-guidelines/espacomp-medication-adherence-reporting-guideline-emerge/).

Authors’ responses: Thank you. We have revised the sentence and it now reads: “We followed the Strengthening of the Reporting of Observational Studies in Epidemiology (STROBE) guidelines and the ESPACOMP Medication Adherence Reporting Guidelines (EMERGE) to report the study findings.”

2. It should be described more clearly which predictors you included in the multivariate regression.

Authors’ responses: We have described the predictors in the multivariate regression analysis. The new sentence reads: “In the multivariate regression analysis, we included tribe, marital status, level of income, current alcohol consumption, duration on ART, health education on ART related side effects, and HIV status disclosure.”

3. The original revision should be supplemented by these suggestions. 

Authors’ responses: Thank you. We have supplemented the original version of the manuscript with the aforementioned suggestions.

---

## [Decision Letter · Decision Letter 2]

10 Nov 2020

Alcohol consumption increases non-adherence to ART among people living with HIV enrolled to the community-based care model in rural northern Uganda

PONE-D-20-22867R2

Dear Dr. Izudi,

We’re pleased to inform you that your manuscript has been judged scientifically suitable for publication and will be formally accepted for publication once it meets all outstanding technical requirements.

Kind regards,

Tim Mathes

Academic Editor

PLOS ONE

Additional Editor Comments (optional):

Reviewers' comments:

Reviewer's Responses to Questions

**Comments to the Author**

1. If the authors have adequately addressed your comments raised in a previous round of review and you feel that this manuscript is now acceptable for publication, you may indicate that here to bypass the “Comments to the Author” section, enter your conflict of interest statement in the “Confidential to Editor” section, and submit your "Accept" recommendation.

Reviewer #1: All comments have been addressed

Reviewer #2: All comments have been addressed

2. Is the manuscript technically sound, and do the data support the conclusions?

Reviewer #1: Yes

Reviewer #2: Yes

3. Has the statistical analysis been performed appropriately and rigorously? 

Reviewer #1: Yes

Reviewer #2: Yes

4. Have the authors made all data underlying the findings in their manuscript fully available?

Reviewer #1: Yes

Reviewer #2: Yes

5. Is the manuscript presented in an intelligible fashion and written in standard English?

Reviewer #1: Yes

Reviewer #2: Yes

6. Review Comments to the Author

Reviewer #1: Dear Editor

The authors have attended to all my concerns. Well done This will contribute significantly to the improvement in HIV treatment outcomes. I do give them a go ahead.

Reviewer #2: 1. Introduction section; the second sentence (line 61-62) should be revised for better clarity.

2. Ethical issues under the methods section, was the informed consent witnessed in both situations of participants who could and could not write/read?

7. PLOS authors have the option to publish the peer review history of their article (what does this mean?). If published, this will include your full peer review and any attached files.

Reviewer #1: **Yes: **Dr M Madhombiro

Reviewer #2: **Yes: **Dominic Bukenya

---

## [Editor Report · Acceptance letter]

13 Nov 2020

PONE-D-20-22867R2 

Alcohol consumption increases non-adherence to ART among people living with HIV enrolled to the community-based care model in rural northern Uganda 

Dear Dr. Izudi:

I'm pleased to inform you that your manuscript has been deemed suitable for publication in PLOS ONE. Congratulations! Your manuscript is now with our production department. 

Kind regards, 

on behalf of

Dr. Tim Mathes 

Academic Editor

PLOS ONE